# The Current Situation and Future Direction of Nanoparticles Lubricant Additives in China

**Kun Han** [1], **Yujuan Zhang** [1], **Ningning Song** [1], **Laigui Yu** [1], **Pingyu Zhang** [1], **Zhijun Zhang** [1], **Lei Qian** [2] **and Shengmao Zhang** [1,*]

1    Engineering Research Center for Nanomaterials, Henan University, Kaifeng 475000, China
2    Shenzhen Research Institute of Beijing Institute of Technology, Shenzhen 518057, China
*    Correspondence: zsm@vip.henu.edu.cn

**Abstract:** Nanoparticles as lubricant additives demonstrate powerful friction reduction and antiwear properties and are potential alternatives to traditional additives in line with green and environmentally friendly requirements. However, the vast majority of currently available research focuses on the tribological properties of various nanoparticles in base oils at laboratory, which has a large gap with their application in engineering. To cope with the rapid economic and industrial development in China, there is a need to improve the tribological properties of nanoparticles. This paper highlights the current status and development trend of nanoparticles as lubricant additives in China. The factors influencing the tribological properties of nanoparticles, such as their composition, particle size and morphology, as well as the base stocks and their combination with other additives, are summarized. Furthermore, the research progress in the lubrication mechanism of nanoparticles is discussed, and the issues concerning the application of nanoparticles as lubricant additives as well as their future directions are discussed. This review is expected to provide an impetus to guide the design of high-performance, fully formulated lubricant systems containing nanoparticles as the lubricant additive.

**Keywords:** nanoparticle; nanoscale lubricant additive; tribological properties; current situation; future direction





## 1. Introduction

Lubricants consist of base oils and additives, of which additives as the essence of high-performance lubricants play key roles in friction reduction, antiwear and antioxidation abilities and load-carrying capacity, as well as the reliability and operational life of mechanical devices under harsh working conditions. The dramatic development of modern industry in China continuously requires that machinery and equipment transform into large, precise and intelligent ones, and the operating environment of the mechanical moving parts increasingly becomes complex and harsh. As a result, traditional lubricant additives are unable to meet the needs of modern industrial development. Nanoadditives, an emerging class of lubricant additives, are superior to traditional lubricant additives in that they not only have superior friction reduction and antiwear properties but also have unique self-healing capability [1,2].

The researches on nanoadditives in China can be traced back to the 1990s, when the subject of nanomaterial tribology was established [3]. Since then, a variety of research at home and abroad has been dedicated to relevant research in this field, which is contributive to enriching and promoting the research on nanomaterial tribology [4–9].

Viewing the widespread industrial application of nanoadditives in our country, this paper attempts to summarize the significant contributions made by our group and other domestic research groups, covering the effects of the component, particle size and morphology of nanoparticles on their tribological properties as well as their interactions with

base oils and/or other additives. It also talks about the currently existing problems and development trends of nanoadditives.

## 2. Effect of Nanoadditive Composition on Tribological Properties

Usually, nanoparticles refer to materials with a particle size of less than 100 nm in at least one dimension in three-dimensional space, and this small size allows them to easily enter the frictional contact zone and efficiently exert excellent tribological properties [10]. Nanoadditives can be classified by composition into metal, oxide, sulfide or carbon and its derivatives as well as rare earth compounds and others, and various nanoadditives usually rely on unique physicochemical properties to deliver excellent friction-reducing and antiwear properties. Table 1 provides a summary of the classification of nanoparticles with respect to their components.

**Table 1.** Summary of various nanoparticles.

| Component | Nanoparticles | Diameter (nm) | Content (wt%) | Decreasing Degree of COF (%) | Decreasing Degree of WSD (%) | Decreasing Degree of WR (%) | Ref. |
|---|---|---|---|---|---|---|---|
| Metal | Cu | 5 | 0.5 | 37 | / | 90 | [11–15] |
| | Ag | 6–7 | 2.0 | 10 | / | 10 | [16,17] |
| | Bi | 40 | 0.1 | 69 | 37 | / | [18,19] |
| | W | 30–60 | 0.5 | 30 | 19 | / | [20] |
| | Al | 65 | 0.5 | 20 | 30 | / | [21] |
| | Ni | 80–120 | 0.1 | 29 | 39 | / | [22,23] |
| | Ga | $286 \pm 21$ | 0.17 | 39 | / | 93 | [24,25] |
| | Sn | 30–60 | 1.0 | 50 | 62 | / | [26] |
| Oxide | $Al_2O_3$ | 78 | 0.1 | 18 | 42 | / | [27] |
| | $ZrO_2$ | 6–7 | 2.0 | 8 | / | 20 | [28,29] |
| | $Fe_3O_4$ | 45–50 | 1.5 | 58 | 13 | / | [30] |
| | CuO | 7–15 | 0.5 | −5 | 22 | / | [31–33] |
| | $SiO_2$ | 35 | 0.1 | 15 | −3 | / | [19,34,35] |
| | $TiO_2$ | 30 | 0.1 | 15 | 3 | / | [35,36] |
| | $SnO_2$ | 20 | 0.1 | 4 | −9 | / | [35] |
| | ZnO | 4 | 1.2 | 10 | 31 | / | [37,38] |
| Sulfide | CuS | / | 1.0 | 31 | / | 81 | [39–41] |
| | $WS_2$ | 20–60 | 1.0 | 27 | / | 85 | [42–44] |
| | $MoS_2$ | 50–100 | 0.5 | 37 | 35 | / | [45,46] |
| Carbon and its derivative | GP | 195–422 | 0.08 | 16 | / | 26 | [47–52] |
| | CNTs | 10–20 | 0.02 | 24 | 6.6 | / | [53,54] |
| | CQDs | 2.66 | 1.0 | 62 | 89 | / | [55] |
| | Diamond | 110 | 0.2 | 17 | 25 | / | [34,56,57] |
| Rare earth compound | $CeO_2$ | 300–600 | 2.0 | 18 | 16 | / | [58] |
| | $La(ReO_4)_3$ | / | 0.5 | 28 | 30 | / | [59] |
| | Mixed rare earth naphthenate | 20–30 | 3.0 | 12 | 55 | / | [60] |
| Other | $Fe_3O_4@MoS_2$ | 600–800 | 1.0 | 44 | / | 20 | [61] |
| | $SiO_2@Cu$ | 694 | 1.0 | 32 | / | 67 | [62] |
| | $SiO_2@MoS_2$ | 683 | 1.0 | 32 | / | 58 | [62] |
| | $Cu@MoS_2$ | 8–13 | 0.5 | 38 | 29 | / | [63] |
| | $MoS_2@CNT$ | 80 | 1.0 | 33 | / | 98 | [64] |
| | $MoS_2@GP$ | 80 | 1.0 | 20 | / | 98 | [64] |
| | $MoS_2@C_{60}$ | 100 | 1.0 | 25 | / | 96 | [64] |
| | $Mn_3O_4/GP$ | 25 | 0.03 | 35 | / | 76 | [65,66] |
| | $WS_2/GP$ | 100 | 0.02−0.04 | 70 | / | 66 | [67] |
| | $TiO_2/BP$ | 300–500 | 0.01 | 26 | 13 | / | [68] |

### 2.1. Metal

Due to their small size effect, metal nanoparticles have a very high surface energy and can be well adsorbed on a rubbed surface during friction. Furthermore, tribochemical reactions may occur under shear stress, resulting in a tribofilm with good friction reduction and antiwear properties. However, metal nanoparticles are chemically unstable and are prone to oxidation in air. These drawbacks, fortunately, can be overcome by

surface capping with oil-soluble alkyl chains, thereby preventing the oxidation of metal nanoparticles and improving their dispersion stability in lubricants [69]. A large number of studies demonstrate that copper nanoparticles are widely adaptable to different lubrication systems and can significantly improve the tribological properties of base oils as well as the contact fatigue life of mechanical components [70–72]. Zhang et al. [73] prepared alkyl phosphorothioate acid (DDP)-modified copper nanoparticle (NPCuDDP) and used it as the nanoadditive for a diamond-like carbon (DLC)/PAO tribosystem (PAO refers to poly-alpha olefin). Unlike conventional small-molecule additives, NPCuDDP is widely adaptable to DLC coating and can significantly reduce the friction coefficient and wear rate of DLC/PAO tribosystems, which is due to the formation of the tribofilm with a low shear strength via tribochemical reactions.

Namely, the Cu nanoparticle undergoes tribochemical reactions to generate oxides and sulfates under shear stress, while the dopants in the DLC coating also participate in tribochemical reactions, thereby contributing to the formation of the tribofilm. The formation of the tribofilm is schematically illustrated in Figure 1.

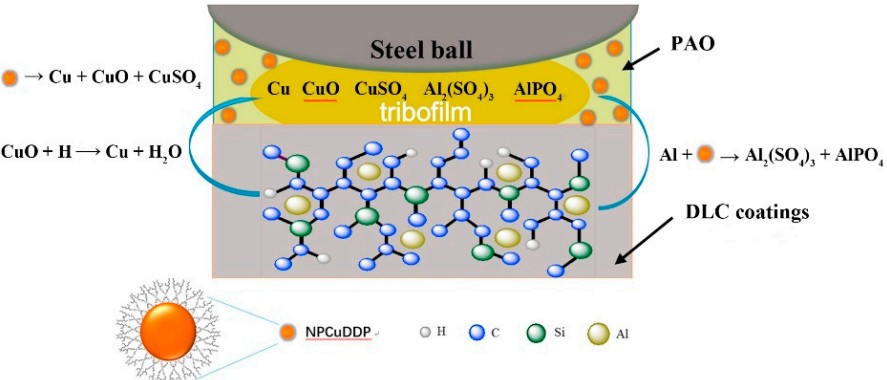

**Figure 1.** Schematic diagram illustrating the formation of the tribofilm of DLC/PAO solid–liquid lubricating system in the presence of NPCuDDP [73].

In addition to Cu nanoadditives, Ni nanoparticle with magnetism as well as catalytic activity has also received much attention [74,75]. Hu et al. [76] used oleylamine-modified Ni nanoparticle (OA-Ni) to form an in situ carbon-based tribofilm with excellent friction reduction and antiwear properties as well as a certain degree of load-carrying capacity, and they suggest that the excellent tribological properties of the OA-Ni nanoparticle is attributes to its catalytic activity towards the formation of a graphitized carbon layer on the rubbed metal surfaces, as shown in Figure 2.

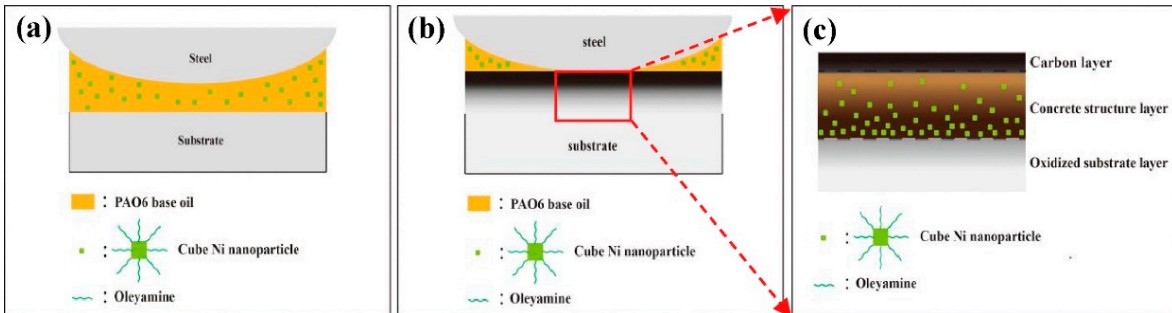

**Figure 2.** Formation mechanism of tribo-film from OA-Ni nanoparticle. (**a**) Stable dispersion of Ni nanoparticles in oil, (**b**) under the catalysis of Ni nanoparticles, PAO6 molecules carbonize around them to form amorphous carbon phase, forming the concrete structure of nickel nanocrystals embedded in the amorphous carbon phase, (**c**) is a partially enlarged schematic of the multilayer composite tribofilm in (**b**) [76].

*2.2. Oxide*

Oxide nanoparticles are advantageous over metal nanoparticles due to their better chemical stability [77]. Nevertheless, it is still a challenge to efficiently and easily fabricate oxide nanoparticles with homogeneous particle size and controllable morphology, which limits their production on a large scale and their application in industry as well. Nevertheless, a range of published research demonstrates that oxide nanoparticles might represent one of the future directions of nanoadditives.

Zhang et al. [78] prepared DDP-modified ZnO nanoparticle (ZODDP) from zinc dialkyldithiophosphate (ZDDP) using a one-step method and investigated the tribological properties of the as-prepared ZODDP for a steel–aluminum contact. As shown in Figure 3, ZODDP has excellent tribological properties and can reduce the friction coefficient and wear rate by 10% and 70%, respectively, thanks to the formation of ZnO deposited film, which is encouraging for overcoming the poor tribological performance of ZDDP towards steel–aluminum contact. This, in combination with the significant reduction in the sulfur and phosphorus contents of ZODDP, could be of particular significance for the development of a new generation of nanoadditives with improved environmental acceptance.

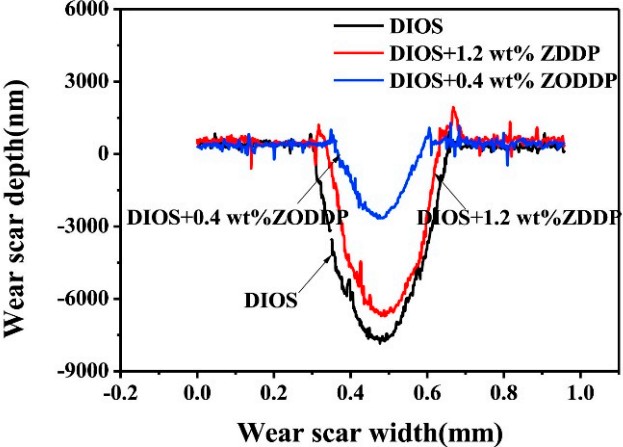

**Figure 3.** Two-dimensional (2D) morphologies of worn surfaces of aluminum disks by using three-dimensional (3D) profilometer (Bruker Contour GT-K) [78].

Huang et al. [79] prepared mesoporous $SiO_2$ (denoted as MSN-loaded T512) with an average particle size of 50 nm to support commercial antioxidants and further evaluated its tribological and antioxidant properties in base oils. Their findings indicate that $SiO_2$ nanoparticle not only has a certain degree of friction-reducing and antiwear abilities but also has good antioxidation ability. As shown in Figure 4, the slow release of the antioxidant prolongs its residence time in the lubricating oil and finally serves a good antioxidant function.

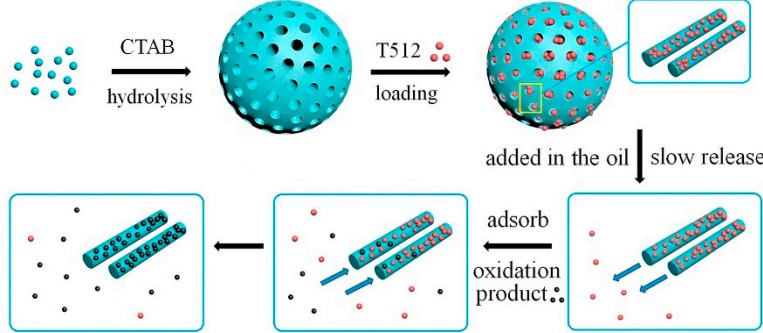

**Figure 4.** Schematic illustration for the preparation of MSNs-loaded T512 and its antioxidant mechanism [79].

### 2.3. Sulfide

Sulfides mainly include $MoS_2$, $WS_2$ and CuS, and sulfur can promote the reaction between nanoparticles and the friction interface to afford high-performance tribofilm, thereby exerting friction-reducing and antiwear effects. However, in keeping with the implementation of increasingly strict laws and regulations for environmental protection, the release of gaseous sulfide in the preparation of metal sulfide is highly limited, and the establishment of the low-temperature green synthesis method for fabricating metal sulfide is still a challenge. Jiang et al. [80] synthesized oil-soluble $WS_2$ nanoadditive using green liquid-phase pyrolysis in the presence of $(NH_4)_2WS_2O_2$ as the precursor and oleic acid as the modifier. As shown in Figure 5, the as-synthesized oil-soluble $WS_2$ nanoadditive can significantly improve the tribological properties of the base oil over a wide temperature range, and it could be a potential alternative to ZDDP, thanks to its excellent friction reduction and antiwear properties superior to those of ZDDP.

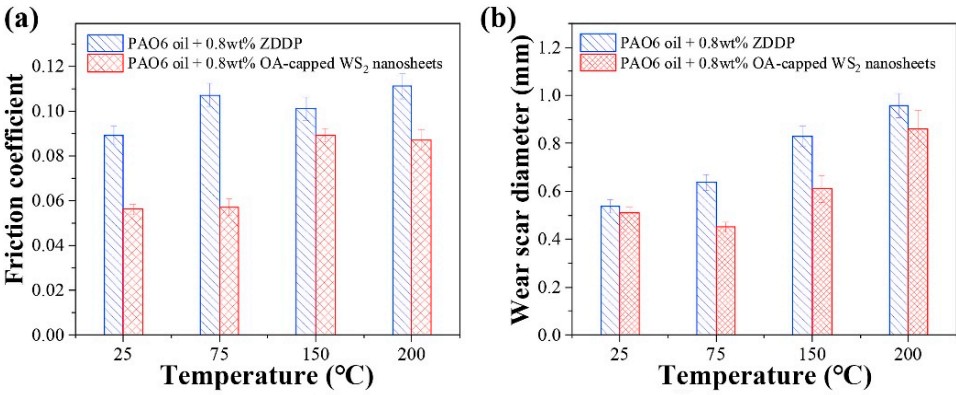

**Figure 5.** (**a**) Friction coefficient and (**b**) wear scar diameter of $WS_2$ and ZDDP in base oil [80].

A spherical $MoS_2$ nanoparticle synthesized by Xu et al. [81] exhibits good friction reduction and antiwear effects in DIOS base oil, which is attributed to the formation of the tribofilm and the adsorption film of $MoS_2$-DIOS on the rubbed metal surfaces. Chen et al. [82] demonstrated that the presence of ultra-thin $MoS_2$ nanosheets could significantly improve the load-carrying capacity of base oils depending on the fracture strength of $MoS_2$. As the Hertzian pressure in the frictional contact region is below the fracture strength of $MoS_2$, the $MoS_2$ nanosheets can prevent direct contact between the rubbed surfaces, which inhibits the occurrence of wear.

### 2.4. Carbon and Its Derivatives

Common carbon and its derivatives, including nanodiamond, carbon nanotubes (CNTs), graphene (GP) and fullerene, have received extensive attention, thanks to their excellent chemical stability, self-lubricity and good mechanical properties [83,84]. A study by Chu et al. [85] from Taiwan Province of China showed that the introduction of 3% (volume fraction) of nanodiamond could minimize the occurrence of friction and wear. Additionally, the addition of carbon nanoparticle prepared by the one-pot method in PAO6 base oil could reduce the friction coefficient and wear spot diameter of the sliding pair by 47% and 30%, respectively, which demonstrates that the as-prepared carbon nanoparticle could be a potential nanoadditive [86]. However, many carbon-based materials suffer from complex preparation, high cost and inhomogeneous product size and are currently unavailable for large-scale industrial production and application [87–89]. In this sense, the search for an efficient and low-cost as well as size-controllable synthesis method for nanoadditives is one of the major future directions worth special attention.

Carbon quantum dots (CQDs), consisting of carbon nuclei and surface groups and usually less than 10 nm in diameter, can combine their surface groups with modifiers possessing different functions, which not only contributes to improving their dispersion stability but also endows them with additional properties compared with conventional

nanodiamond. Ye et al. [90] synthesized diphenylamine-modified multifunctional CQDs with photoluminescence and good antioxidant ability as well as friction-reducing and antiwear properties at low temperature. As an additive to polyethylene glycol (PEG) base oil, CQDs with a concentration of 1% (mass fraction) in the PEG base oil exhibit the best friction-reducing and antiwear properties. The presence of the diphenylamine structure gives it the ability to scavenge free radicals, and the antioxidant capacity of CQDs is positively dependent on their concentration.

Since its first report in 2004, GP has been extensively investigated in a wide range of areas such as electronics and mechanics [91], and in recent years, its applications have gradually expanded into the field of nanoadditives. A variety of GP-based materials have been derived due to their excellent mechanical strength, thermal conductivity and oxidative corrosion resistance as well as good lubricity. Li et al. [92] used isopropyl triisostearyl titanate to surface-modify graphene oxide (GO) and obtained T-GO with excellent dispersion stability. The as-prepared T-GO can significantly improve the extreme pressure properties of the base oil and reduce the friction coefficient and wear rate by 50% and 20%, respectively.

*2.5. Rare Earth Compound*

Rare earth elements are important raw materials for national defense and high-tech industries [93]. China, the country with the largest rare earth reserve in the world, is at the forefront of the research on rare earth. In recent years, nanoadditives containing rare earth elements have attracted extensive attention, largely because the nanoparticles made from rare earth compounds with high chemical activity and favorable adsorption capability often exhibit desired friction-reducing and antiwear functions. Jin et al. [94] investigated the tribological properties of oil-soluble lanthanum fluoroborate (La(BF$_4$)$_3$-OA) in polar base oils, and they found that the active elements La, B and F could promote the formation of the protective tribofilm on the friction interface, thereby exerting excellent friction-reduction and antiwear properties. Wu et al. [95] synthesized oleylamine-modified CeO$_2$ (denoted as OA-CeO$_2$) nanoparticle by a one-pot method and evaluated its tribological properties. They found that a low concentration of the as-prepared OA-CeO$_2$ nanoparticle can catalyze the oxidation of the friction subsurface to form an antiwear protective film with iron oxide as the main component. At the same time, a high concentration of CeO$_2$ nanoparticle can form a deposited film with high load-carrying capacity, which also contributes to improving the friction-reducing and antiwear properties of the lubricant (Figure 6). Some studies show that rare earth elements are biologically toxic and their stockpile levels are continuously decreasing along with their applications in a wide range of high technology, which will inevitably harm the use of rare earth compounds as lubricant additives in the future [96,97]. This reminds us that the recycling of rare earth elements from waste materials, and their use in the preparation of environmentally friendly nanoadditive could be worth special attention.

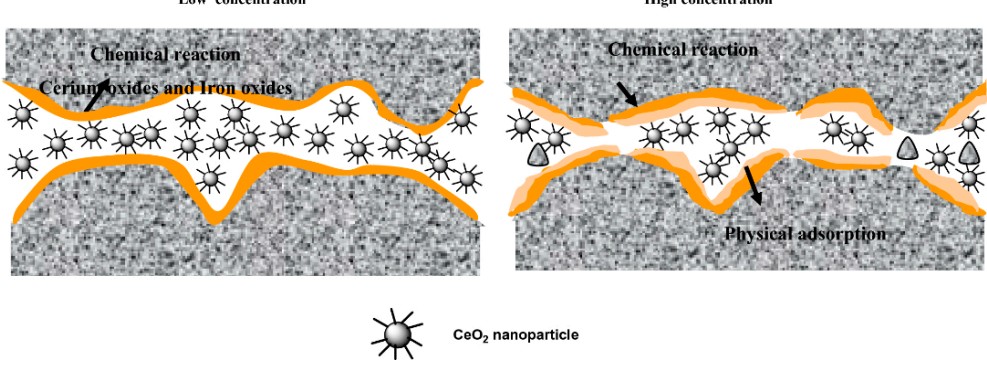

**Figure 6.** Schematic illustration of the antiwear mechanism of OA-modified CeO$_2$ nanoparticle under different concentrations [95].

### 2.6. The Others

Today, many efforts are being made to design and prepare nanohybrids as lubricant additives. Compared with a single nanoparticle, nanohybrids combine the characteristics of over two kinds of nanoparticles, which is favorable for achieving a synergistic triboeffect and thereby more efficiently reducing friction and wear. For example, core–shell-structured nanoparticles are used extensively in the catalytic and electromagnetic fields, and the differences in the physicochemical properties of the core and shell can be utilized to benefit tribological properties [98]. Zhang et al. [99] prepared carbon-coated magnesium silicate hydroxide (MSH@C) by hydrothermal method and found that MSH@C is more effective than single MSH in reducing friction and wear, which is because the core–shell-structured nanoparticle participates in a tribochemical reaction to generate a loose tribofilm with a highly ordered structure. As shown in Figure 7, the core–shell structure is separated during the friction process: The shell is adsorbed onto the rubbed surface to further undergo phase transition under shear stress, while the core moves out of the contact region to carry the load.

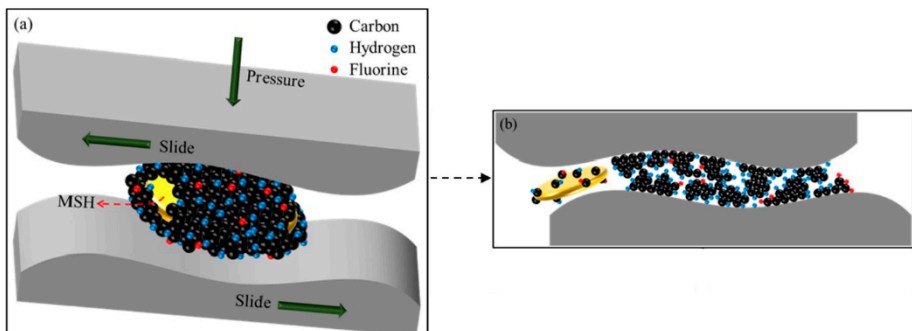

**Figure 7.** (**a**) The MSH@C undergoes core–shell separation by compression and shear. (**b**) The shell layer adheres to the newly worn surface by physical adsorption or mechanical embedding, while the thin MSH core slips out of the sliding contact interface [99].

For conventional 2D-layered material such as GP, the weak interlayer interactions allow it to slip during friction, thereby reducing friction. However, under extreme conditions, GP cannot provide effective antiwear protection, and in this case, it often needs to combine GP with nanoparticle [100]. Gan et al. [101] prepared ionic liquid-modified GO/Cu nanocomposite and found that the as-prepared nanocomposite as the lubricant additive of PEG can effectively reduce friction and wear (by 40% and 47%).

In addition to nanohybrids, black phosphorus (BP) [102,103] and metal–organic frameworks (MOFs) [104] as nanoadditives have also attracted some attention. BP is nontoxic and thermodynamically stable, and it is widely used in semi-conductive industry (field-effect transistor), biomedicine and catalysis [105]. However, the presence of the solitary pairs of electrons on the surface of BP makes it highly susceptible to reacting with oxygen and water in air, which is harmful to its structure stability [106]. This requires that BP with a 2D-layered structure be treated to avoid premature failure; otherwise, it would be inapplicable to nanoadditives for reducing friction and wear. Tang et al. [107] used oleic acid to modify BP and achieved a macroscopic superlubricity of steel/steel contact with the assistance of the as-obtained BP. The reason lies in that the surface-capped BP forms an adsorbed film in the contact region during friction, and its lone pair of electrons further binds the lubricant oleic acid and catalyzes the degradation of oleic acid to form amorphous carbon, thereby impeding the direct contact between the steel surfaces.

MOFs offer significant advantages in gas storage and separation, biosensing and catalysis, as well as drug delivery [108,109]; and the multi-selectivity of organic linkers is favorable for increasing the dispersion stability of MOFs in oil [110,111], which could help to expand their application in nanoadditives. Wu et al. [112] synthesized zirconium-based MOFs (Zr-MOFs) using DDP as the modifier, and the as-synthesized Zr-MOFs exhibited

excellent dispersion stability in base oil and could reduce friction and wear by more than 50%. At the same time, the Zr-MOFs with a high specific surface area can significantly increase the oxidation induction period of the base oil, and their oxidation resistance is improved after the modification by DDP.

## 3. Effect of Size of Nanoadditive on Tribological Properties

Nanoparticles of different sizes have varying ease of entry into the frictional contact zone under the same test condition, which inevitably leads to discrepancies in tribological properties. Smaller nanoparticles will not only enter the frictional contact zone faster but also exhibit improved dispersion stability and oil transmission capacity, allowing for better judgment of oil quality. Nanoadditives will certainly develop towards smaller particle size in the future [113–117]. However, comparative studies on the tribological properties of nanoparticles of different particle sizes are scarce, and different test conditions can be selective for the optimum particle size. Xu et al. [118] prepared calcium carbonate nanoparticles (CCNPs) of different particle size, where the small CCNPs have the best friction reduction and antiwear properties at high load and low frequency, while the large size CCNPs have the greatest performance at high frequency. In addition, nanoparticles can mend surface damage caused by wear through the deposition effect. Various sizes of nanoparticles are embedded in the surface protrusions to differing degrees, and small nanoparticles that can be fully embedded have the optimal antiwear ability [119].

For laminar nanoparticles, friction depends on the number of layers, and an increase in the number of layers can reduce friction [120]. Ci et al. [121] studied the differences in the tribological properties of hexagonal boron nitride (h-BN) at differing thicknesses. They found that h–BN of a medium thickness has the best friction reduction and antiwear properties, which is because h-BN with a moderate thickness can sustain interlayer sliding during the friction process and can form a 150 nm thick tribofilm.

In addition, the length and diameter of some nanomaterials will also determine their access to the contact area during the friction process. Ye et al. [122] observed that CNTs with short lengths (0.5~2 μm) and medium diameters (10~20 nm) have the best friction reduction and antiwear properties, due to their rolling bearing and surface mending effects between the contact areas.

## 4. Effect of Morphology of Nanoadditive on Tribological Properties

Sphere-, sheet-, onion- and nanotube-like nanoparticles are widely used in tribology, and their lubrication mechanisms and tribological properties are highly dependent on morphology [123]. Various nanoparticles with different morphologies are summarized in Table 2, and the progress in the mechanistic studies will be described later. Usually, it is necessary to artificially manipulate the morphology of nanoparticles so that they can be well applied in engineering, and the increased difficulty and cost of preparation can diminish the benefits of friction reduction and antiwear properties [124]. Therefore, it is imperative to establish cost-effective production methods of nanoparticles; otherwise, the commercialization of nanoadditives will be infeasible.

**Table 2.** Summary of nanoparticles with different morphologies.

| Morphology | Nanoparticles | Diameter (nm) | Content (wt%) | Decreasing Degree of COF (%) | Decreasing Degree of WSD (%) | Decreasing Degree of WR (%) | Ref. |
|---|---|---|---|---|---|---|---|
| Sphere | PDA@Cu | 200 | 0.4 | 45 | / | 97 | [125] |
| | $Fe_3O_4$@$MoS_2$ | 500 | 0.5 | 17 | / | 34 | [126] |
| | CNSs | 40–60 | 0.025 | 48 | 14 | / | [127–131] |
| | CNSs-PEI | 200–600 | 0.3 | 28 | / | 42 | [132] |
| | $MoS_2$ | 20 | 0.5 | 37 | 35 | / | [133] |
| | $MoS_2$/$TiO_2$ | 67 | 1.0 | 20 | 33 | / | [134] |

**Table 2.** *Cont.*

| Morphology | Nanoparticles | Diameter (nm) | Content (wt%) | Decreasing Degree of COF (%) | Decreasing Degree of WSD (%) | Decreasing Degree of WR (%) | Ref. |
|---|---|---|---|---|---|---|---|
| | $MoS_2$ | 400 | 0.06 | 28 | 23 | / | [135] |
| | $Ti_3C_2T_x/MoS_2$ | / | 0.3 | 39 | / | 85 | [136] |
| | Ag/BP | 200–400 | 0.075 | 73 | / | 92 | [137] |
| | LDH | 50–140 | 1.0 | 17 | 30 | / | [138] |
| | LDH/GO | / | 0.2 | 67 | / | 97 | [139] |
| Sheet | LDH/$MoS_2$ | / | 0.2 | 67 | / | 98 | [139] |
| | BN | 200–500 | 0.06 | 35 | 35 | / | [140–143] |
| | $SiO_2$–B–N–GO | 50–100 | 0.15 | 24 | 47 | / | [144] |
| | OAMBN/Cus | / | 0.2 | 27 | 25 | / | [145] |
| | GP | 600 | 0.075 | 27 | 43 | / | [146] |
| | COFs | / | 0.05 | 49 | / | 95 | [147,148] |
| | Bentonite | / | 1.0 | 48 | / | 50 | [149,150] |
| Onion | IF-$WS_2$ | 100–200 | 0.25 | 27 | 43 | 87 | [151] |
| | Candle soot particles | 30–50 | 0.3 | 14 | / | 39 | [152] |
| | $NiFe_2O_4$/OLFs | 30 | / | 73 | / | / | [153] |
| | MWCNTs | / | 0.01 | 8 | / | 91 | [154] |
| | Ag/MWCNTs | / | 0.18 | 36 | 32 | / | [155] |
| Tube | Cu/PDA/CNTs | / | 0.2 | 34 | 24 | / | [156] |
| | Halloysite | / | 0.6 | 24 | / | 28 | [157] |
| | CNTs/$MoS_2$ | / | 3.0 | 44 | 34 | / | [158] |
| | Ni/MWCNT | / | 0.2 | 44 | / | 56 | [159] |

*4.1. Sphere*

Spherical nanoparticles might act as rolling bearings in the frictional contact area to prevent direct contact between the rubbed surfaces, eventually transforming sliding friction into rolling friction [160–162]. Carbon nanospheres (CNSs) are typical representatives of spherical nanoparticles with the advantages of being green and chemically inert and having excellent friction-reducing and antiwear properties [163]. After many years of development, CNSs with mature preparation methods and low cost are suitable for largesacle industrial production [164–168]. Ye et al. [169] said that N and P co-doped CNSs can reduce friction and wear by 60% and 90%, respectively, due to the formation of a protective film on the rubbed surface, and the dual-doped CNSs retain excellent stability even under harsh conditions.

In addition to carbon-containing materials, spherical oxide nanoparticles with excellent mechanical properties have also received widespread attention. Li et al. [170] prepared a $ZrO_2/SiO_2$ nanocomposite that reduced the coefficient of friction by 16% at low concentrations. Zhou et al. [171] designed and synthesized hollow $SiO_2$@$TiO_2$ spherical materials that provided a versatile and green nanoadditive that not only reduces the coefficient of friction and wear rate by 40% and 50% but also rapidly degrades more than 50% of the used base oil within 80 min.

*4.2. Sheet*

Common sheet or 2D-layered nanoparticles include GP, BP, $MoS_2$, BN and others. In contrast to spherical nanoparticles, the weak interactions between the layers allow sheet-like materials to slip during friction, thereby reducing the coefficient of friction. In addition, the strong film-forming capacity [172] of 2D-layered materials, in combination with their polishing effect [173] and mending effect [174], makes them advantageous potential nanoadditives. This section will focus on the application of BN and a new type of sheet-like nanoparticle (2D transition metal carbide, nitride and carbonitride (MXene)) in lubricants.

BN has high chemical inertness and mechanical thermal stability and is widely used in the field of high-temperature solid lubrication [175]. In recent years, BN has been known as a lubricant additive with promising potential in reducing friction and wear; its use in the field of solid lubrication, however, is constrained by its poor dispersion stability

in lubricants. To deal with this issue, Wang et al. [176] used amino-containing silane coupling agents to surface-functionalize h-BN, which in combination with the reaction between 4-carboxyphenyl boronic acid and amino group generates a sheet-like nanoadditive (CPBA-BNNSs) with excellent dispersion stability in lubricant base oil. Compared with the unmodified h-BN, the CPBA-BNNSs exhibit superior tribological properties even at lower additions.

Layered double hydroxide (LDH) is widely used in catalysis, electrode and nuclear element capture and biomaterial [177–179]. Recently, there has been considerable interest from researchers in LDH as an aqueous or oil-based nanoadditive, and numerous studies show that LDH has excellent tribological properties [180–183]. Wang et al. [184] achieved a combination of catalysis and wear reduction by the calcination of Ni-Al LDH in air. The nickel oxide formed after calcination catalyzes the carbonization of the base oil to form carbon chips, thereby effectively avoiding the direct contact between the rubbed surfaces. In turn, the relatively thick tribofilm formed by LDH ensures extremely low wear at high contact pressures.

MXenes are emerging 2D materials with important applications in energy storage, shielding against electromagnetic interference and sensing [185]. The presence of surface groups such as -OH, -O- and -F gives MXenes favorable polarity [186], which extends their application as nanoadditives. Differing from conventional 2D materials, MXenes can provide continuous lubrication at any thickness, and their tribological properties are independent of the interlayer interactions [187]. Yi et al. [188] achieved macroscopic superlubricity with $Ti_3C_2T_x$ MXene nanosheets in glycerol, and the coefficients of friction and wear rate are reduced by more than 95% at lower Hertzian contact pressures. This is because the tribofilm of MXene-containing nanosheets significantly reduces the shear stress under boundary lubrication, while the glycerol layer formed with the nanosheets through hydrogen bonding adsorption can reduce the shear stress under elastic fluid lubrication. As a result, there is a synergistic lubrication effect between MXene and glycerol, as shown in Figure 8. Yi et al. further investigated the macroscopic superlubrication of $Mo_2CT_x$ MXene nanosheets in ionic liquids [189]. Their experimental results show that a composite tribofilm containing molybdenum oxide and phosphorus oxide, in conjunction with the low shear strength between the MXene nanosheet layers, achieve macroscopic superlubricity at a maximum Hertzian contact pressure of above 1.4 GPa.

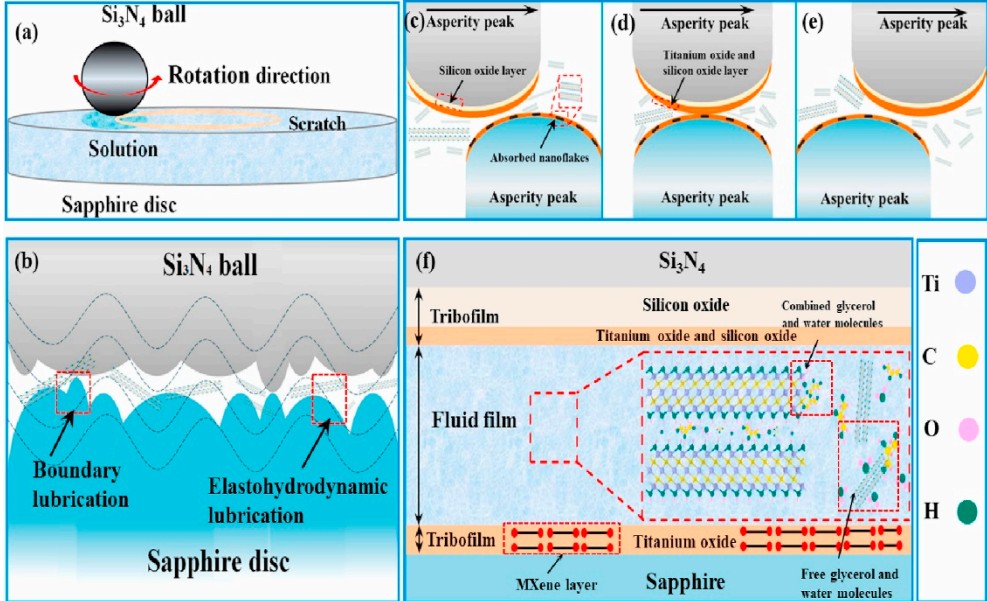

**Figure 8.** (**a**) Schematic diagram of the superlubricity system, (**b**) schematic contact zone, (**c**–**e**) contact zone of the boundary lubrication, and (**f**) schematic illustration of elastohydrodynamic lubrication [188].

Furthermore, covalent organic backbones such as bentonite clays and their composites are also of great interest to researchers. They will not be discussed in detail here due to the limited space available. In one word, 2D-layered materials with unique advantages have found applications in various fields, and they could give rise to novel nanoadditives with promising potential in engineering.

### 4.3. Onion

Onion nanoparticles with concentric multilayer spherical structure are thought to have outstanding tribological properties, especially under extreme conditions such as high temperature and high loading [190–192]. Unlike sheet and spherical nanoparticles, onion nanoparticles can undergo rolling friction rather than sliding at low contact pressures, while their exfoliated lamellae with a low shear strength can provide sliding friction at high-contact pressures [193]. Meanwhile, the absence of surface dangling bonds is favorable for ensuring the chemical inertness and reducing the adhesion to friction subsurface of onion-like nanoparticles, which is conducive to the occurrence of rolling friction [8].

He et al. [194] prepared onion-like carbon nanoparticle by a short period of gas bursting, and the as-obtained onion-like carbon nanoparticle as the lubricant additive greatly reduced friction and wear because the exfoliated GP nanosheets can form a protective film with desired friction reduction and antiwear properties. Luo et al. [195] obtained an onion structure by the spontaneous combination of $MoS_2$ nanosheets through laser irradiation. The as-obtained onion structure $MoS_2$, with a minimal amount of dangling bond, has greatly reduced surface energy as well as excellent antioxidant behavior and extreme pressure properties. Ouyang et al. [196] investigated the tribological properties of onion $WS_2$ and flake talc nanoparticles as lubricant nanoadditives. They found that the onion $WS_2$ with a high load-carrying capacity as well as mending effect for the friction subsurface reduced the Hertzian contact pressure, while the interlayer sliding of talc nanoparticle helps to further reduce the shear stress via a synergistic effect.

### 4.4. Nanotube

Nanotubular materials with varying aspect ratios tend to roll upon entering the frictional contact area [197]. However, strong intermolecular forces inevitably cause the nanotubes to form aggregates whose rolling bearing effect is much weaker than that of spherical nanoparticles. It is argued that rolling is not a reasonable lubrication mechanism of nanotubes due to their special characteristics such as susceptibility to deformation and wide distribution and the presence of structural defects. Instead, the exfoliation and deformation of nanotubes in combination with the formation of thick boundary lubrication film are mainly responsible for the desired friction reduction and antiwear properties of nanotubular materials [198,199].

CNTs are the most widely studied class of tubular materials [200]. The formation of carbon films under dry friction gives them excellent self-lubricating performance, and their self-lubricity is positively correlated with their content [201]. As oil-based nanoadditives, CNTs often need to be surface modified in order to improve their dispersion stability in the lubricant base oil and hinder their agglomeration therein. Bai et al. [202] prepared a supramolecular gel lubricant by dispersing acid-treated CNTs in a 500SN base oil containing a gelling agent. As shown in Figure 9, the 3D mesh structure formed by the gelling agent in the base oil can effectively improve the dispersion stability of CNTs, and the addition of a small amount of CNTs is favorable for improving the friction reduction and antiwear properties. Gong et al. [203] improved the dispersion stability of CNTs in oil by covalent modification with polymeric aryl phosphates. They found that the protective tribofilm formed on the surface of the steel disc allows the composite to retain excellent friction reduction and antiwear properties at high temperatures.

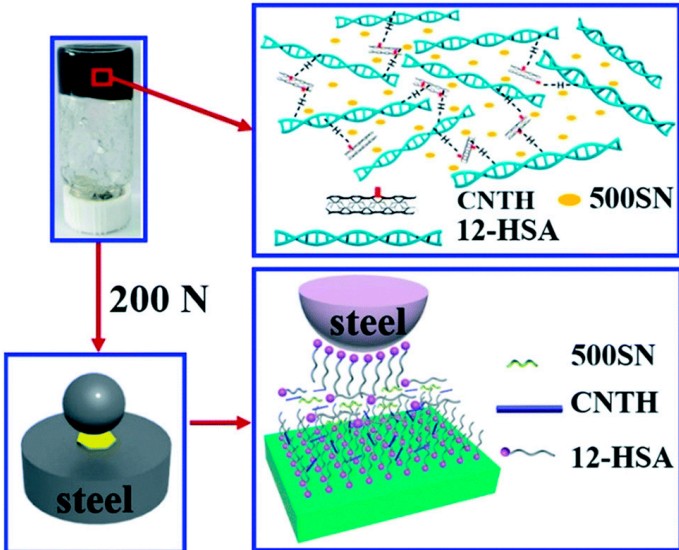

**Figure 9.** Schematic diagram of lubrication mechanism of CNTs in 500SN base oil [202].

## 5. Lubrication Mechanisms

### 5.1. Ball Bearing Effect

The spherical or sphere-like nanoparticles, with rigid structure, act like ball bearings when they enter the frictional contact area, transforming sliding friction into rolling combined with sliding friction [204]. In this way, the direct contact between the rubbed surfaces is prevented, and the occurrence of extreme situations such as jamming are greatly hindered. At the same time, the high-performance nanoparticles with active surface groups are confined to the frictional contact area owing to their desired embedding stability [205], which is favorable for them to exert a ball-bearing effect. Duan et al. [206] suggested that rolling rather than sliding friction exists between the rubbed interfaces when $ZnO/Al_2O_3$ composite nanoparticles are used as lubricant additives. Similarly, talc/carbon sphere composite nanoparticle can transform sliding friction into rolling friction when it enters the friction interval, and there is a synergistic effect between the talc and carbon spheres [207].

### 5.2. Mending Effect

Wear can contribute to the appearance of cracks and micro-pits on worn surface, and nanoparticles entering the friction contact area can be deposited within the defects to mend the worn surface efficiently, thereby compensating for the wear mass loss. This mending effect of nanoadditive can help to reduce the surface roughness, thereby improving fuel efficiency and extending the service life. For example, surface-modified cellulose nanocrystals exhibit good dispersion stability in lubricant base oil and can mend the wear marks on rubbed surfaces [208]. Similarly, the ionic liquid-modified CNTs exert a ball-bearing effect during friction as well as mending and polishing effects, which could account for their excellent friction reduction and antiwear properties [209].

### 5.3. Polishing Effect

Mechanical components are regularly ground to enhance surface accuracy and reduce surface roughness. Similarly, nanoparticles in lubricants can act as abrasive grains to fill and polish surfaces once introduced into the frictional contact area, thereby reducing the mass loss caused by friction and wear. Usually, the polishing effect of nanoparticles highly depends on their mechanical properties. For example, a surface treated with nanodiamond nanofluid possessing high hardness, modulus of elasticity and compressive strength can achieve the best surface smoothness, in contrast with those treated with commonly seen nanoparticles such as $Al_2O_3$, $SiO_2$ and $MoS_2$ [210].

*5.4. Formation of Tribofilm*

Small nanoparticles with a high specific surface area have high adsorption capacity and reactivity in the boundary lubrication zone. It has been shown that organic–inorganic hybrid nanoparticles can form a protective film on rubbed surfaces via organic chain adsorption thereon under mild condition, and the adsorption film becomes thinner until it ruptures at an extended friction duration [211]. At this point, the nanoparticles are deposited in situ on the friction subsurface under shear stress, forming a tribofilm with desired extreme pressure and antiwear properties. In contrast to traditional small-molecule organic additives, nanoparticles form tribofilms mainly through the welding of inorganic nuclei, adhesion or reaction with the surface and do not consume or corrode the friction substrate. When the film formation rate and the wear rate reach a dynamic equilibrium, no more wear occurs, whereas when the film formation rate increases further, even negative wear occurs in association with the mass augment of the worn surface.

*5.5. Extension of Tribomechanism of Nanoadditive*

The friction reduction and antiwear properties of nanoparticles as lubricants have long been considered to be related to the four lubrication mechanisms mentioned above. With the continuous development of tribology, a large number of nanoparticles is being used as lubricant nanoadditives, thus enabling the classical lubrication mechanisms to be continuously supplemented. For instance, graphite- and graphene-based protective films formed on metal substrates have excellent tribological properties, and their mechanistic models have been continuously optimized with development [212]. Within the frictional contact region, sheet nanoparticles can transform the interfacial shear of the friction subsurface into the intralayer shear of nanoparticles, which in association with weak interlayer interactions contributes to greatly reducing interfacial friction and wear [213–218]. As a complement to the mending effect and tribofilm formation, friction-sintering mechanism is increasingly receiving attention [219–221]. Namely, in the presence of normal load and shear stress, nanoparticles as the lubricant additives are diffused into the friction subsurface and rapidly densified without reaching the sintering temperature, thereby forming friction-sintered film with excellent friction reduction and antiwear properties [222].

## 6. Effect of Other Additives on the Tribological Properties of Nanoparticles

Typical lubricant additives include friction reducer, antiwear agent, cleaner, dispersant and viscosity modifier. In practice, a single nanoadditive cannot meet all the performance requirements of the lubricant and needs to be combined with other additives to afford fully formulated high-performance finished lubricant. However, there are often complex interactions between nanoparticles and other additives, which could lead to synergistic or antagonistic triboeffect. For example, the excessive adsorption of the dispersant on the friction subsurface could affect the adhesion of the tribofilm, thus preventing the nanoparticle from properly exerting friction reduction and antiwear properties [223,224]. Another example in this respect is that there is synergistic effect between $MoS_2$ and extreme pressure, antiwear as well as scavenging agents, whereas there is antagonistic effect between $MoS_2$ and dispersant [225]. Currently, few domestic studies are available about the combination of nanoadditives with other additives, and the commercial application of nanoparticles as lubricant additives is still challenging. This, however, should not negate that with the implementation of the dual-carbon target in our country, the study on green and efficient fully formulated lubricants suitable for advanced machinery lubrication will be a popular topic in forthcoming research.

Our group previously investigated the effect of various types of additives on the lubricating performance of the oil-soluble Cu nanoparticle [226]. There are synergistic friction reduction and antagonistic antiwear effects among the dispersant, scavenger, and Cu nanoparticle. There are synergistic friction-reducing and antiwear effects between Cu nanoparticle and antioxidant, while there is an antagonistic antioxidant effect between them. Similarly, a synergistic triboeffect occurs among friction modifier, viscosity modifier

and ZDDP, and a synergistic antioxidant effect occurs between the viscosity modifier and ZDDP. Lei et al. [227] investigated the tribological properties of ZDDP combined with oleylamine-modified $CeO_2$ and found that the synergistic co-adsorption, i.e., the rigid monolayer adsorption of ZDDP and the viscoelastic adsorption of $CeO_2$, increases the nonseizure load.

Ionic liquids can be used as base oils, friction modifiers or antiwear additives [228,229]. The combination of nanoparticle and ionic liquids has also attracted much attention from researchers [230,231]. Li et al. [232] demonstrated that there is a synergistic triboeffect between Mo nanoparticle and the ionic liquid of 2-mercaptobenzothiazolate. In PEG base oil, the hybrid additive has the optimum tribological properties, which is because the anion of the ionic liquids and Mo nanoparticle participate in tribochemical reactions to generate in situ $MoS_2$ tribofilm on the friction subsurface. Qu et al. [233] synthesized the ionic liquid of 1-ethoxycarbonylmethyl$-$3-methylimidazolium tetrafluoroborate, and they found a synergistic lubrication effect between the as-synthesized ionic liquids and Cu nanoparticle due to the formation of tribofilm consisting of copper oxide and boron trifluoride on the rubbed surface.

## 7. Effect of Different Base Oils on the Tribological Properties of Nanoparticles

For a more convenient description, this article divides the commonly used base oils into two categories: the low-polarity base oils represented by mineral oils and polyalphaolefins and the high-polarity base oils represented by diisooctyl sebacate. The adsorption of polar base oil on the rubbed surfaces prevents the deposition of nanoparticles thereon and the ball bearing effect as well, which hinders the friction reduction and antiwear effects of the nanoparticles. At the same time, nanoparticles incorporated in the oil film might damage the integrity of the oil film, thereby aggravating wear [234]. In general, there are discrepancies in the tribological properties of nanoparticles as lubricant additives in separate base oils. A good example in this respect is that Cu nanoparticle in synthetic ester base oil can increase wear by more than seven times, while in mineral oils it can significantly reduce wear [235]. Relevant studies in this area, however, still remain at the initial stage all over our country, and mastering the chaotropic performance diversities of various nanoparticles with respect to diurnal base oils has important implications for the commercial application of nanoadditives as well as for compounding novel high-performance lubricants.

## 8. Conclusions

Nanoparticles of numerous components, sizes and morphology have been attracting extensive attention as lubricant additives. They often have unique advantages over traditional small-molecule additives, due to the small size effect and good access to surface-functionalization. Their largescale application in industry, however, lags and awaits the establishment of more cost-effective preparation methods based on further research and exploration.

(1) Domestic researchers have mainly concentrated on research into the application of traditional nanoparticles such as metals and oxides. As the country with the largest rare earth content, a large number of rare earth materials are used in various medium- and high-end equipment manufacturing industries every year, which generates a large amount of waste. The use of waste containing rare earth elements to synthesize nanoadditives with excellent tribological properties is useful not only for the reuse of resources but also for environmental protection. At the same time, the rise of emerging materials including composite nanomaterials has also shown that single nanoparticles cannot meet the needs of industry, so the development of high-performance nanocomposites should be vigorously pursued based on the original research. Ultimately, the organic combination of small-scale laboratory experiments, pilot plants and large-scale preparation will promote the rapid development of nanoadditives in China.

(2) Although domestic researchers have conducted some research on onion- and tube-like nanoparticles, their difficulty of preparation and the economic cost inevitably restrict their development. In contrast, sphere- and sheet-like nanoparticles are accepted for more products. In the future, we suggest concentrating on the synthesis and preparation of ultra-small nanoparticles, as well as their application, and achieve large-scale production.

(3) China has always been a pioneer in environmental protection, and with the introduction of laws and regulatory requirements related to environmental protection, bio-based lubricants with desirable biodegradability will dominate the market in the future. At this stage, the research on nanoadditives by Chinese researchers is mainly limited to mineral base oils. Compared with mineral oils, bio-based lubricants have different molecular structures and even diametric physicochemical properties, which refers to diverse requirements for additives. Therefore, it is imperative to promote the basic study of nanoadditives suitable for bio-based lubricating base oil and to explore their multiple interactions in relation to their formulation rules.

(4) In the context of domestic industrial development and the current status of nanoadditives applications. With a view to the priority research directions in the future, we suggest focusing on the quantitative relationship between the structure, component and tribological properties of nanoadditives as well as their compatibility with other lubricant additives such as antioxidant, viscosity index improver, cleaner and dispersant, and the conformational relationships between the morphology–microstructure and tribological properties of nanoadditives, the correlation between their surface chemistry and tribological properties and the pathways to tuning their tribological properties based on the surface molecular design are worth special attention. Furthermore, it is urgent to establish macro-preparation technology for nanoadditives and to explore in depth the design and synthesis of self-dispersed nanoadditives as well as nanoadditives suitable for biodegradable base oils in order to promote their applications in industry.

**Author Contributions:** Writing—Original draft: K.H.; Data curation: Y.Z., N.S., L.Y., P.Z., Z.Z. and L.Q.; Writing—Review & editing: S.Z. All authors have read and agreed to the published version of the manuscript.

**Funding:** This research was funded by the National Natural Science Foundation of China (51875172; 52105180); Zhongyuan Science and Technology Innovation Leadership Program (214200510024); and Zhujiang Talent Program (2016LJ06C621).

**Institutional Review Board Statement:** Not applicable.

**Informed Consent Statement:** Not applicable.

**Data Availability Statement:** Not applicable.

**Conflicts of Interest:** The authors declare no conflict of interest.

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
