# Peer review of "The Current Situation and Future Direction of Nanoparticles Lubricant Additives in China"

_lubricants, doi:10.3390/lubricants10110312_

Round 1

Reviewer 1 Report

1-The title of the review suggests the economic aspect of China in the lubrication, but it is different in the scientific content and the focus is not on China enough.

2-I detect plagiarism is very little (20%) with (Exclude bibliography On) is acceptable.

Title:

Current situation and future direction of Nanoparticles lubricant  additives in China

 Abstract:

It is better for the Abstract to contain the following:

1st Background

2nd Aim

3rd Methods

4th Results and Discussion

5th Conclusions

The content above must be implicitly adhered to when writing the abstract.

- Remove Fig. 2

- In Figure 4 , It is best to display  best Figure of the required information ( just one figure). , And with the name of the technique used for measurement (by using......)

- Rewrite title of Figure 6

- No need Fig.7 (Remove Figure 7)

- In Fig.9  Choose only two figures  and adding all details,  with an explanation of a and b

- Remove Fig. 10

- Insert Table No. 1 after the last paragraph (The others) page 11 , before paragraph ( 3. Effect of size ......... properties.)

- No need Fig.11 (Remove Figure 11)

- No need Fig.14 and 15 (Remove Figure 14 and 15)

- Insert Table No. 2 after the last paragraph (4.3. Nanotube) page 17 , before paragraph (5. Lubrication mechanisms.)

 Notes:

-2.1. Metal , 2.2. Oxide

It is necessary to be brief in writing a review of the concepts of researchers

(You should synthesize the researchers’ ideas in short and comprehensive texts)

(You should stay away from writing repeating the concepts of researchers in the same subject)

 2.4. Carbon and its derivatives

(You should avoid repetition of writing in the concepts of researchers in the same topic)

and collected into one concept

2.6. The others

(You should avoid repetition of writing in the concepts of researchers in the same topic)

and collected into one concept

The text should be shortening (removing) by 25% of total text. While preserving the scientific content of the text

3. Effect of size of nanoadditive on tribological properties

The text should be shortening (removing) by 25% of total text. While preserving the scientific content of the text

4.1. Sphere, 4.2. Sheet

You should avoid repeating the concepts of researchers in the same subject

The text should be shortening (removing) by 25% of total text.

6. Effect of other additives on the tribological properties of nanoparticles

You should avoid repeating the concepts of researchers in the same subject

The text should be shortening (removing) by 25% of total text.

  8. Conclusions

All paragraphs (points) (must include information related to the country of China) (from researchers results) and future (your recommendations)

- Paragraphs (first - second - third) are rejected

- Paragraphs (fourth-fifth) must include information related to the country of China

- It is better to rely on the information of Table No. 1 in writing the conclusions (and mention the current and future situation represented by the recommendations for the State of China)

(Metal , Oxide ,Sulfide ,Carbon and its derivative ,Rare earth compound ,Other)

The methods available in China - the availability of raw materials - the availability of manufacturing methods in China - the best way…… etc.

- It is also better to rely on the information of Table No. 2 in writing the conclusions (and mention the current and future situation represented by the recommendations for the State of China) (Sphere, Sheet, Onion, Tube )

The methods available in China - the availability of raw materials - the availability of manufacturing methods in China - the best way…… etc.

Notes

It is better to include information related to the economic aspect of China in a new table or add it in Tables 1 and 2

Author Response

Reviewer #1:

1-The title of the review suggests the economic aspect of China in the lubrication, but it is different in the scientific content and the focus is not on China enough.

I detect plagiarism is very little (20%) with (Exclude bibliography On) is acceptable.

Response: We are grateful for your comments. We checked and revised the content of the paper, introduced and summarized the relevant work in China, and reflected it in the paper.

2-Title: Current situation and future direction of Nanoparticles lubricant additives in China

Response: Thank you very much for your suggestion and we have revised the title.

3-Abstract: It is better for the Abstract to contain the following:

1st Background

2nd Aim

3rd Methods

4th Results and Discussion

5th Conclusions

The content above must be implicitly adhered to when writing the abstract.

Modification: We are very grateful for the comments of the reviewers. We have added the research aims, removed some of the research backgrounds and expressed the overall content of the article in a concise and condensed way in the abstract. Finally, a graphical abstract has been added to the abstract.

4-Remove Fig. 2

- In Figure 4, It is best to display best Figure of the required information (just one figure). And with the name of the technique used for measurement (by using......)

- Rewrite title of Figure 6

- No need Fig.7 (Remove Figure 7)

- In Fig.9 Choose only two figures and adding all details, with an explanation of a and b

- Remove Fig. 10

- Insert Table No. 1 after the last paragraph (The others) page 11, before paragraph (3. Effect of size ......... properties.)

- No need Fig.11 (Remove Figure 11)

- No need Fig.14 and 15 (Remove Figure 14 and 15)

- Insert Table No. 2 after the last paragraph (4.3. Nanotube) page 17, before paragraph (5. Lubrication mechanisms.)

Modification: Thank you for your suggestions. We have removed Fig. 2, 7, 10, 11, 14 and 15, in addition to adjusting the position of Table 1 and 2. Moreover, only one representative image has been selected for Fig. 4 and the name of the technique and instrument used for that image has been added to the caption. Changes were made to the caption of Fig. 6. Two of the three images in Fig. 9 have been selected as a combination of (a) and (b) and their contents have been explained.

5-Notes:

-2.1. Metal, 2.2. Oxide

It is necessary to be brief in writing a review of the concepts of researchers

(You should synthesize the researchers’ ideas in short and comprehensive texts)

(You should stay away from writing repeating the concepts of researchers in the same subject)

2.4. Carbon and its derivatives

(You should avoid repetition of writing in the concepts of researchers in the same topic)

and collected into one concept

2.6. The others

(You should avoid repetition of writing in the concepts of researchers in the same topic)

and collected into one concept

The text should be shortening (removing) by 25% of total text. While preserving the scientific content of the text

  1. Effect of size of nanoadditive on tribological properties

The text should be shortening (removing) by 25% of total text. While preserving the scientific content of the text

4.1. Sphere, 4.2. Sheet

You should avoid repeating the concepts of researchers in the same subject

The text should be shortening (removing) by 25% of total text.

  1. Effect of other additives on the tribological properties of nanoparticles

You should avoid repeating the concepts of researchers in the same subject

The text should be shortening (removing) by 25% of total text.

Modification: We are grateful for your comments. We have collated and revised the similar research work in the relevant sub-sections you mentioned and have removed around 25% of these.

6-Conclusions

All paragraphs (points) (must include information related to the country of China) (from researchers results) and future (your recommendations)

- Paragraphs (first - second - third) are rejected

- Paragraphs (fourth-fifth) must include information related to the country of China

- It is better to rely on the information of Table No. 1 in writing the conclusions (and mention the current and future situation represented by the recommendations for the State of China)

(Metal, Oxide, Sulfide, Carbon and its derivative, Rare earth compound, Other)

The methods available in China - the availability of raw materials - the availability of manufacturing methods in China - the best way…… etc.

- It is also better to rely on the information of Table No. 2 in writing the conclusions (and mention the current and future situation represented by the recommendations for the State of China) (Sphere, Sheet, Onion, Tube)

The methods available in China - the availability of raw materials - the availability of manufacturing methods in China - the best way…… etc.

Modification: Thank you for your suggestions. We have removed (1), (2) and (3) and enriched (4) and (5) to include information about China. Conclusion (1) has been reformulated based on Table 1, highlighting that China should make use of the abundance of rare earth resources to effectively use the waste from the production of medium and high-end rare earth products to produce nanoparticles containing rare earth elements, which is in line with green requirements and to develop nanocomposites. Conclusion (2) is re-proposed based on the contents of Table 2, suggesting that nanoparticles should move towards ultra-small sizes in the future, and that spherical and sheet-like nanoparticles should be applied more practically due to their simplicity of preparation compared to tube and onion-like shapes.

7-Notes

It is better to include information related to the economic aspect of China in a new table or add it in Tables 1 and 2

Response: Thanks for your advice. But I am sorry that I was unable to complete the tables on the Chinese economy. Due to the paucity of news data and related research papers, and in the interests of scientific rigor, I am incapable of drawing reliable conclusions using a small amount of data.

Reviewer 2 Report

I've read the review with a great interest. Nevertheless, I've got a feeling that the paper possesses a number of demerits the editors are mainly to judge.

Minor

1. This pretends to be a scientific paper rather than journalism. Therefore, too personified clauses should be avoided. E.g., the passage in lines 45-48 MUST BE replaced with the following:

"The researches on nanoadditives in China can be traced back to 1990s when the subject of nanomaterial tribology was established [3]."

2. In lines 60-63 the authors dwell on what the nanosize ist. It is not correct. At least one of the dimensions should be smaller than 100 nm. The other dimensions can be larger. The authors should stick to the point. This has a significant statistical (thermodynamical) ground. The authors can better consult here:  

  • DOI: 
  • 10.1016/B978-0-32-390543-5.00022-0

Major

3. The paper is devoted to advancement of the Chinese scientific community in nanoparticle additive experimental research. It is interesting but the question is whether it is justified for a paper in the international journal.

4. The paper contains a number of sub-sections (spherical NPs, sheet-, onion-, nanotube-shaped etc.; lubrication effects as to ball bearing, polishing etc.) which are the well-known aspects and have been comprehensively described in other reviews. The contribution of the authors to the relevant considerations and discussions seems to be tiny.   

Author Response

Reviewer #2:

I've read the review with a great interest. Nevertheless, I've got a feeling that the paper possesses a number of demerits the editors are mainly to judge.

Response: Thank you for your suggestion. I have revised my paper in order to try to minimize the shortcomings.

Minor

  1. This pretends to be a scientific paper rather than journalism. Therefore, too personified clauses should be avoided. E.g., the passage in lines 45-48 MUST BE replaced with the following:

"The researches on nanoadditives in China can be traced back to 1990s when the subject of nanomaterial tribology was established [3]."

Modification: We are very grateful for the question of the reviewer. We have amended the sentences in the relevant paragraphs with your comments.

  1. In lines 60-63 the authors dwell on what the nanosize ist. It is not correct. At least one of the dimensions should be smaller than 100 nm. The other dimensions can be larger. The authors should stick to the point. This has a significant statistical (thermodynamical) ground. The authors can better consult here:

DOI:10.1016/B978-0-32-390543-5.00022-0

Modification: Thanks for your advice. I have carefully reviewed the paper that you suggested to study and it has benefited me immensely. I have made modifications to the concept of nanoparticles.

Major

  1. The paper is devoted to advancement of the Chinese scientific community in nanoparticle additive experimental research. It is interesting but the question is whether it is justified for a paper in the international journal.

Response: Thank you for your suggestion. China is the second largest economy in the world with the largest market and industrialization, so it is vital to promote the development of the domestic lubrication field. This paper is dedicated to promoting the development of domestic and international cooperation in the field of lubrication by presenting the current status of Chinese research in the field of nanoadditives and the direction of development, so it is necessary to publish in an international journal. This will not only make the world recognize China but will also make domestic researchers aware of their inadequacy.

  1. The paper contains a number of sub-sections (spherical NPs, sheet-, onion-, nanotube-shaped etc.; lubrication effects as to ball bearing, polishing etc.) which are the well-known aspects and have been comprehensively described in other reviews. The contribution of the authors to the relevant considerations and discussions seems to be tiny.

Response: We are very grateful for the question of the reviewer. This paper focuses on the current state of research and future trends in the field of nanoadditives in China, which is in line with the mainstream research direction worldwide, so what is explored in this paper has some similarities to other reviews. However, in addition to introducing traditional nanoadditives, this paper also focuses on the application of novel nanoparticles in lubricants, such as MXene and MOF materials. Furthermore, beyond the four traditional lubrication mechanisms, this paper summarises advances in emerging mechanisms such as weak intra-layer interactions and friction sintering. It is helpful for the development of tribological research.

Round 2

Reviewer 1 Report

Don’t delete Table 1. Summary of various nanoparticles

Line 716 delete As shown in Table 1,

Line 728 delete As shown in Table 2,

Line 744 delete we and Rewrite (we suggest to)

Line 40 delete review

Author Response

Reviewer 1: Review Report (Round 2)

1 Don’t delete Table 1. Summary of various nanoparticles.

Response: We are very grateful for the comments of the reviewers. We have not deleted Table 1. Summary of various nanoparticles. We have inserted Table 1 after the last paragraph (The others), before paragraph (3. Effect of size......... properties.) as you previously suggested.

2 Line 716 delete As shown in Table 1,

Line 728 delete As shown in Table 2,

Line 744 delete we and Rewrite (we suggest to)

Line 40 delete review

Response: Thank you very much for your suggestion, we deleted the appropriate content.

Reviewer 2 Report

In general, I am satisfied with improvements introduced into the manuscript.

What is related to my major remarks in the previous review (#3 and #4), the Editorial Board is to finally decide.

Meanwhile, I have to ask the authors to take into account the following. There is a very fresh paper published recently:
https://doi.org/10.1007/s40544-022-0685-7

This is a comprehensive review, which should be taken into account and necessarily cited.

There, there are references which could be cited as well as they are affine to what the authors underline in their response to my earlier comments: "... this paper summarises advances in emerging mechanisms such as weak intra-layer interactions"

Thus, one has to take into account and to possibly cite a number of works in section 2.4.1 Carbon-based materials. Among them are work [140] and a number of works cited on page 1455  (look there for the clause "The lubrication mechanism is mainly attributed to the fact that graphene or graphene derivatives enter the contact area to form a friction protective film, which reduces wear and reduces friction significantly due to the weak interlayer force of graphene nanosheets.", please).

Author Response

In general, I am satisfied with improvements introduced into the manuscript.

What is related to my major remarks in the previous review (#3 and #4), the Editorial Board is to finally decide.

Meanwhile, I have to ask the authors to take into account the following. There is a very fresh paper published recently:

https://doi.org/10.1007/s40544-022-0685-7

This is a comprehensive review, which should be taken into account and necessarily cited.

There, there are references which could be cited as well as they are affine to what the authors underline in their response to my earlier comments: "... this paper summarises advances in emerging mechanisms such as weak intra-layer interactions"

Thus, one has to take into account and to possibly cite a number of works in section 2.4.1 Carbon-based materials. Among them are work [140] and a number of works cited on page 1455 (look there for the clause "The lubrication mechanism is mainly attributed to the fact that graphene or graphene derivatives enter the contact area to form a friction protective film, which reduces wear and reduces friction significantly due to the weak interlayer force of graphene nanosheets.", please).

Response: We are very grateful for the suggestion of the reviewer. We promptly read the paper to which you refer, and we benefited from the richness of its content. In particular, the content of carbon-based materials is an important reference for this review, so we take some of the references and explain them. The cited references are as follows:

  1. Vigdorowitsch, M.; Ostrikov, V.V.; Sazonov, S.N.; Safonov, V.V.; Orobinsky, V.I. How Carbon-Based Nanosheets Protect: Mechanistic Models. Lett. 2021, 69, 1-9,doi:10.1007/s11249-021-01478-y.
  2. Meng, Y.; Xu, J.; Ma, L.; Jin, Z.; Prakash, B.; Ma, T.; Wang, W. A review of advances in tribology in 2020–2021. Friction2022, 10, 1443-1595, doi:10.1007/s40544-022-0685-7.
  3. Wu, J.; Yin, X.; Mu, L.; Feng, X.; Lu, X.; Shi, Y. Hollow IF-MoS2/r-GO Nanocomposite Filled Polyimide Coating with Improved Mechanical, Thermal and Tribological Properties. Coatings2020, 11, 25, doi:10.3390/coatings11010025.
  4. Zhao, B.; Yu, X.; Liu, Y.; Yang, L.; Zhang, Z.; Zhang, B. Frictional characteristics of heterostructure film composed of graphene and H-BN with the consideration of defects. Int.2021, 153, 106607, doi:10.1016/j.triboint.2020.106607.
  5. Chen, H.; Ba, Z.; Qiao, D.; Feng, D.; Song, Z.; Zhang, J. Study on the tribological properties of graphene oxide composite films by self-assembly. Int.2020, 151, 106533, doi:10.1016/j.triboint.2020.106533.
  6. Rejhon, M.; Lavini, F.; Khosravi, A.; Shestopalov, M.; Kunc, J.; Tosatti, E.; Riedo, E. Relation between interfacial shear and friction force in 2D materials. Nanotechnol.2022, doi:10.1038/s41565-022-01237-7.